# Exploratory mass cytometry analysis reveals immunophenotypes of cancer treatment-related pneumonitis

**Toyoshi Yanagihara[1,2]\*[†], Kentaro Hata[1][†], Keisuke Matsubara[3], Kazufumi Kunimura[3], Kunihiro Suzuki[1], Kazuya Tsubouchi[1], Satoshi Ikegame[1], Yoshihiro Baba[4], Yoshinori Fukui[3], Isamu Okamoto[1]**

[1]Department of Respiratory Medicine, Graduate School of Medical Sciences, Kyushu University, Fukuoka, Japan; [2]Department of Respiratory Medicine, NHO Fukuoka National Hospital, Fukuoka, Japan; [3]Division of Immunogenetics, Department of Immunobiology and Neuroscience, Medical Institute of Bioregulation, Kyushu University, Fukuoka, Japan; [4]Division of Immunology and Genome Biology, Department of Molecular Genetics, Medical Institute of Bioregulation, Kyushu University, Fukuoka, Japan

**\*For correspondence:**
toyoshi.yana@gmail.com

[†]These authors contributed equally to this work and share the first authorship

**Abstract** Anticancer treatments can result in various adverse effects, including infections due to immune suppression/dysregulation and drug-induced toxicity in the lung. One of the major opportunistic infections is *Pneumocystis jirovecii* pneumonia (PCP), which can cause severe respiratory complications and high mortality rates. Cytotoxic drugs and immune-checkpoint inhibitors (ICIs) can induce interstitial lung diseases (ILDs). Nonetheless, the differentiation of these diseases can be difficult, and the pathogenic mechanisms of such diseases are not yet fully understood. To better comprehend the immunophenotypes, we conducted an exploratory mass cytometry analysis of immune cell subsets in bronchoalveolar lavage fluid from patients with PCP, cytotoxic drug-induced ILD (DI-ILD), and ICI-associated ILD (ICI-ILD) using two panels containing 64 markers. In PCP, we observed an expansion of the CD16+ T cell population, with the highest CD16+ T proportion in a fatal case. In ICI-ILD, we found an increase in CD57+ CD8+ T cells expressing immune checkpoints (TIGIT+ LAG3+ TIM-3+ PD-1+), FCRL5+ B cells, and CCR2+ CCR5+ CD14+ monocytes. These findings uncover the diverse immunophenotypes and possible pathomechanisms of cancer treatment-related pneumonitis.

## eLife assessment

This study presents a **valuable** inventory of immune signatures that are correlated with cancer treatment-related pneumonitis. The data were collected and analyzed using validated methodology and can be used as a starting point for further prospective studies. The authors have provided an scRNA-seq analysis with an HD baseline using publicly available dataset and the evidence for their claims is **convincing**.

## Introduction

The advancements in anticancer therapy have revolutionized cancer management and have led to improved patient outcomes. However, these therapies can also lead to various adverse effects, including infections due to immune suppression/dysregulation and drug-induced toxicity, with the lungs being a commonly affected organ (*Conte et al., 2022*; *Morelli et al., 2022*; *Skeoch et al., 2018*).

One of the major opportunistic infections in cancer patients or those receiving immunosuppressive treatment is *Pneumocystis jirovecii* pneumonia (PCP), which can cause severe respiratory complications and high mortality rates (*Thomas and Limper, 2004*; *Asai et al., 2022*; *Apostolopoulou and Fishman, 2022*). However, the precise immune reactions during PCP development and mechanisms of lung injury remain largely unknown.

While both cytotoxic drugs and immune-checkpoint inhibitors (ICIs) can induce interstitial lung diseases (ILDs), the mechanisms underlying these complications may differ between the two classes of drugs. In particular, ICI-associated ILDs are considered a type of immune-related adverse events (irAEs) (*Postow et al., 2018*; *Ando et al., 2021*). Nonetheless, the precise mechanisms that cause drug-induced ILDs are predominantly unidentified. Further, despite various diagnostic approaches, including radiology and molecular testing, there is a need for useful biomarkers to distinguish PCP, cytotoxic drug-induced (DI)-ILD, and ICI-ILD.

Mass cytometry, also known as cytometry by time-of-flight (CyTOF), is a cutting-edge technology that uses inductively coupled plasma mass spectrometry to detect metal ions tagged to antibodies that bind to specific cellular proteins, allowing for a detailed and multidimensional characterization of cellular composition and function (*Spitzer and Nolan, 2016*). This technique provides a unique advantage over traditional flow cytometry, which is limited by spectral overlap and the number of parameters that can be analyzed simultaneously. Mass cytometry has been used to study a variety of biological systems, including immune-mediated diseases, such as cancer and autoimmune conditions (*Matsubara et al., 2021*; *Hata et al., 2023*; *Couloume et al., 2021*; *Chedid et al., 2022*). Through high-throughput analysis of large numbers of single cells, mass cytometry can provide a more comprehensive understanding of the cellular heterogeneity and signaling pathways involved in these diseases, allowing for the identification of potential therapeutic targets and biomarkers.

This study aims to distinguish pulmonary involvement in patients with malignancy undergoing chemotherapy and identify potential biomarkers for DI-ILD, ICI-ILD, and PCP by analyzing bronchoalveolar lavage fluid (BALF) samples with mass cytometry. By characterizing the cellular and molecular changes in BALF from patients with these complications, we aim to improve our understanding of their pathogenesis and identify potential therapeutic targets.

**Table 1.** Characteristics of the study population.

| | PCP | DI-ILD | ICI-ILD |
|---|---|---|---|
| Number | 7 | 9 | 9 |
| Age | 66.9 ± 4.6 | 58.8 ± 19.3 | 71.9 ± 4.5 |
| Male % (n) | 42.8 (3) | 77.8 (7) | 100 (9) |
| Suspected drug/regimen | | VAC (2), FOLFIRI (2), DTX (1), AC (1), GEM (1), TS-1 (1), BEP (1) | Nivolumab (3), pembrolizumab (3), durvalmab (2), atezolizumab (1) |
| CTCAE grade | G2 (5), G3 (1), G5 (1) | G1 (2), G2 (7) | G2 (7), G3 (2) |
| *BALF cell differentiation (%)* | | | |
| Macrophage | 25 ± 8.9 | 40.2 ± 23.5 | 44.4 ± 26.6 |
| Neutrophil | 13.6 ± 29.7 | 1.9 ± 1.3 | 14.4 ± 26.7 |
| Lymphocyte | 58.8 ± 30.5 | 56.2 ± 24.8 | 35.6 ± 23.7 |
| Eosinophil | 0.9 ± 1.1 | 1.7 ± 1.9 | 5.6 ± 8.5 |

Data for age and BALF analysis are presented as means ± SD.

PCP = *Pneumocystis jirovecii* pneumonia; DI-ILD = cytotoxic drug-related interstitial lung disease; ICI-ILD = immune-checkpoint inhibitor-related ILD; BALF = bronchoalveolar lavage fluid; VAC = vincristine, actinomycin D, and cyclophosphamide; FOLFIRI = folinic acid, fluorouracil, and irinotecan hydrochloride; DTX = docetaxel; AC = adriamycin and cyclophosphamide; GEM = gemcitabine; TS-1 = tegafur gimeracil oteracil potassium; BEP = bleomycin, etoposide, and platinum; CTCAE = Common Terminology Criteria evaluated the severity of the disease for Adverse Events grading; grade 1 = asymptomatic; grade 2 = symptomatic; grade 3 = severe symptoms, requiring oxygen therapy; grade 4 = life-threatening respiratory failure; grade 5 = death.

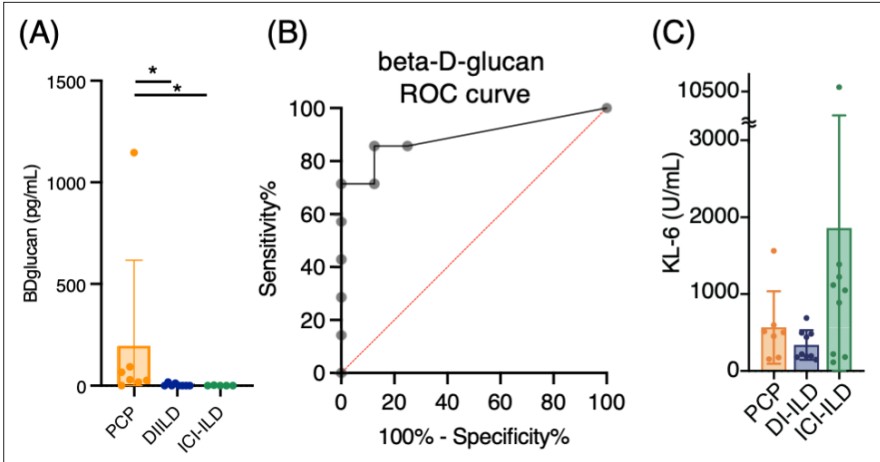

**Figure 1.** Serum levels of β-ᴅ-glucan and KL-6 from patients with *Pneumocystis jirovecii* pneumonia (PCP), cytotoxic drug-related interstitial lung disease (DI-ILD), and immune-checkpoint inhibitor-related ILD (ICI-ILD). (**A**) Serum levels of β-ᴅ-glucan from patients with PCP, DI-ILD, and ICI-ILD. (**B**) A receiver operating characteristic (ROC) curve was constructed to evaluate the diagnostic utility of β-ᴅ-glucan for PCP, with values from the DI-ILD group used as a reference. The area under the curve was determined to be 0.8929. (**C**) Serum levels of KL-6.

## Results

### Patient characteristics and clinical parameters

We analyzed seven cases of PCP, nine of DI-ILD, and nine of ICI-ILD (*Table 1*). One of the PCP cases was fatal. Differential cell counts for BALF revealed lymphocytosis in seven out of seven cases (100%) of PCP as well as in eight out of nine cases (88.9%) of DI-ILD and seven out of nine cases (63.6%) of ICI-ILD when the cutoff for the percentage of lymphocytes was set to >20%. Serum levels of β-ᴅ-glucan were significantly higher in patients with PCP than those with DI-ILD or ICI-ILD (196.5 ± 419.8, 3.7 ± 7.1, 0.6 ± 1.4 pg/mL, respectively) (*Figure 1A*). The receiver operating characteristic (ROC) curve was constructed to evaluate the diagnostic utility of β-ᴅ-glucan for PCP, with values from the DI-ILD group used as a reference (*Figure 1B*). The area under the curve was determined to be 0.8929. When the cutoff value was set to 14.70 pg/mL, the sensitivity and specificity of PCP were 85.71 and 87.50%, and the positive likelihood ratio was 6.86. The diagnostic value of β-ᴅ-glucan was consistent with the previous study (*Tasaka et al., 2007*), where the positive likelihood ratio was 6.57, with a cutoff value of 31.0. The serum levels of KL-6, an indicator of various types of interstitial pneumonitis (*Ishikawa et al., 2012*), tended to be higher in patients with ICI-ILD than PCP or DI-ILD, though insignificant (*Figure 1C*). Four out of seven (57.1%) in PCP, one out of nine (11.1%) in DI-ILD, and six out of nine (66.7%) in ICI-ILD were positive for KL-6 with a cutoff value of 500 U/mL.

### Expansion of CD16$^+$ T cells in BALF from patients with PCP

First, we investigated whether subsets of T cells (gated CD2$^+$ CD3$^+$) differentially existed in PCP, DI-ILD, and ICI-ILD with mass cytometry. T cell lymphocytosis was observed in all groups (PCP: 62.9 ± 17.5%; DI-ILD: 58.3 ± 24.5%; ICI-ILD: 50.4 ± 29.9%) with a higher tendency of CD4/CD8 ratio observed in DI-ILD compared to PCP and ICI-ILD (PCP: 1.13 ± 0.86; DI-ILD: 6.08 ± 6.06; ICI-ILD: 1.9 ± 1.40) (*Figure 2A*). To visualize T cell differentiation within the affected lungs, we generated Uniform Manifold Approximation and Projection (UMAP) plots (*Figure 2B*). The majority of T cells in the BALF displayed either memory or effector phenotypes, with a limited population of naïve T cells (*Figure 2B and C*). Specifically, we observed a tendency of a higher abundance of transient memory/effector memory CD4 T cells and a lower abundance of transient memory/effector memory CD45RA (EMRA) CD8 T cells in DI-ILD compared to the other two groups (*Figure 2C*).

We then utilized the Citrus algorithm to classify T cell subpopulations with varying degrees of abundance in an unsupervised manner by analyzing 31 parameters (*Figure 2D*, *Figure 2—figure supplement 1C and D*). Our analysis identified 30 clusters of T cells, of which 13 were significantly differentiated between the groups. Interestingly, clusters #26299, #26293, and #26278, characterized

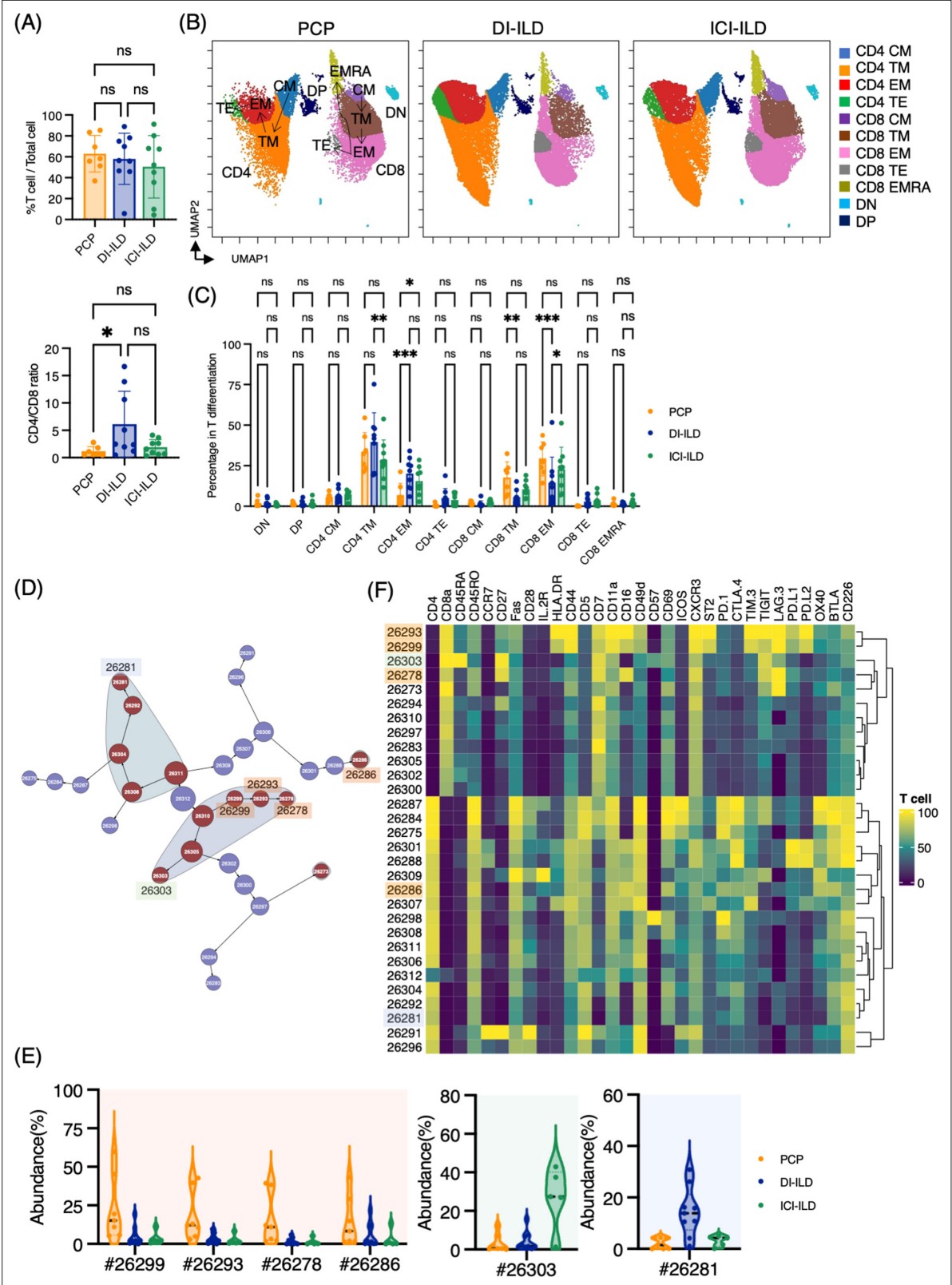

**Figure 2.** Characterization of T cell subsets in bronchoalveolar lavage fluid (BALF) from patients with *Pneumocystis jirovecii* pneumonia (PCP), cytotoxic drug-related interstitial lung disease (DI-ILD), and immune-checkpoint inhibitor-related ILD (ICI-ILD). (**A**) Percentage of T cells (defined as CD2$^+$CD3$^+$) in CD45$^+$ BALF cells and CD4/CD8 ratio in T cells from patients with PCP, DI-ILD, and ICI-ILD. (**B**) Uniform Manifold Approximation and Projection (UMAP) of concatenated samples visualizing the distribution of T cell subpopulations. Central memory (CM) T cells were defined by CCR7$^+$ CD45RO$^+$

*Figure 2 continued on next page*

Figure 2 continued

CD28+ Fas+, transitional memory (TM) by CCR7- CD45RO+ CD28+ Fas+, effector memory (EM) by CCR7- CD45RO+ CD28- Fas+, terminal effector (TE) by CCR7- CD45RO+/- Fas-, and effector memory RA (EMRA) by CCR7- CD45RO- CD45RA+ Fas+/-. Arrows indicate the trajectory of T cell differentiation. DN: CD4- CD8- double negative; DP: CD4+ CD8+ double positive. (C) Percentage of T cell subpopulations. (D) Citrus network tree visualizing the hierarchical relationship of each marker between identified T cell populations gated by CD45+CD2+ CD3+ from PCP (n = 7), DI-ILD (n = 9), and ICI-ILD (n = 5). Clusters with significant differences are represented in red, and those without significant differences in blue. Circle size reflects the number of cells within a given cluster. (E) Citrus-generated violin plots for six representative and differentially regulated populations. Each cluster number (#) corresponds to the number shown in panel (D). (F) Heatmap demonstrates the expression of various markers in different clusters of T cells, as identified through the Citrus analysis. All differences in abundance were significant at a false discovery rate < 0.01.

The online version of this article includes the following figure supplement(s) for figure 2:

**Figure supplement 1.** A T cell gate (CD2+CD3+) and Citrus analysis of T cell populations in bronchoalveolar lavage fluid (BALF) cells from *Pneumocystis jirovecii* pneumonia (PCP), cytotoxic drug-related interstitial lung disease (DI-ILD), and immune-checkpoint inhibitor-related ILD (ICI-ILD).

**Figure supplement 2.** T cell marker expression on the Uniform Manifold Approximation and Projection (UMAP) in *Figure 2B*.

by CD8+ T cells expressing CD16 (FcγRIIIa), and cluster #26286, characterized by CD4+ T cells expressing CD16, were abundant in PCP compared to DI-ILD and ICI-ILD (*Figure 2E and F*). These CD8+ T cells expressing CD16 were also positive for HLA-DR and CXCR3 (*Figure 2F*) but negative for CD14, a marker for monocytes (median CD14 expression of these clusters: 0.0–0.1 vs. 7.4–9.5 in monocyte populations). Cluster #26303, prevalent in ICI-ILD, comprised CD8+ CD57+ TIGIT+ LAG3+ but CD16- subpopulation (*Figure 2E and F*). These cells also expressed PD-1 and TIM-3, although the expression was lower than clusters #26299, #26293, and #26278. These CD8+ T cells expressed CD45RA but no expression of CD45RO and CCR7, indicating EMRA phenotype. Cluster #26281, prevalent in DI-ILD, was marked by CD4+ with low immune-checkpoint expression (PD-1, TIM-3, TIGIT, LAG3, PD-L1, PD-L2, and OX40; *Figure 2E and F*).

## Myeloid cell subpopulations in the lungs of PCP, DI-ILD, and ICI-ILD

Next, we investigated myeloid cell populations (identified as CD3- CD11b+ CD11c+). We first conducted UMAP to see the major myeloid cell populations. The UMAP plot categorized four major subtypes: monocytes, CCR2+ macrophages, alveolar macrophages, and dendritic cells, with no significant difference in PCP, DI-ILD, and ICI-ILD (*Figure 3A and B*). We next utilized the Citrus algorithm to further investigate differently abundant myeloid cell subpopulations by analyzing 18 parameters. Our analysis identified 31 clusters of myeloid cells, of which 17 were significantly differentiated between the groups (*Figure 3C*, *Figure 3—figure supplement 1*). Clusters #8220, #8215, and #8195, prevalent in PCP, were comprised of CD11bhi CD11chi CD64+ CD206+ alveolar macrophages with HLA-DRhi (*Figure 3D and E*). Clusters #8219 and #8197, prevalent in ICI-ILD, were characterized by CD14+ CCR2+ CCR5+ monocyte subpopulations (*Figure 3D and E*). Clusters #8227, #8223, and #8208, prevalent in DI-ILD, were marked by CD11blo CD11clo CD64lo CCR5+ subpopulations (*Figure 3D and E*).

## B cell subpopulations in the lungs of ILDs

In addition to T cells and myeloid cells, we sought to investigate whether there were differential representations of B cells in BALF from PCP, DI-ILD, and ICI-ILD. By utilizing CD45+CD3-CD64- and CD19+ or CD138+ as gating parameters, we were able to detect the presence of B cells and plasma cells. The frequency of B cells/plasma cells was found to be relatively higher in patients with PCP compared to the other two groups, although the proportion of B cells and plasma cells remained low in all groups. (*Figure 4A*). A t-stochastic neighborhood embedding (t-SNE) analysis of 17 parameters among B cells/plasma cells revealed the presence of various B cell subpopulations, including IgD+-naïve B cells, IgM+ B cells, IgG+ B cells, IgA+ B cells, plasmablasts, and plasma cells (*Figure 4B*). Although not significant, higher levels of IgG+ B cells were observed in patients with PCP, while individuals with ICI-ILD exhibited a greater presence of plasma cells (*Figure 4C*). Previous research has suggested that FCRL5+ B cells contribute to the pathogenesis of autoimmune disorders (*Owczarczyk et al., 2020*; *Dement-Brown et al., 2012*). Therefore, we investigated the frequency of these cells in our study and found that FCRL5+ B cells were more abundant in patients with ICI-ILD compared to those with PCP and DI-ILD, suggesting these FCRL5+ B cells may have a role in irAE (8.37 ± 6.74%, 1.37 ± 2.30%, 2.29 ± 3.81%, respectively) (*Figure 4D*).

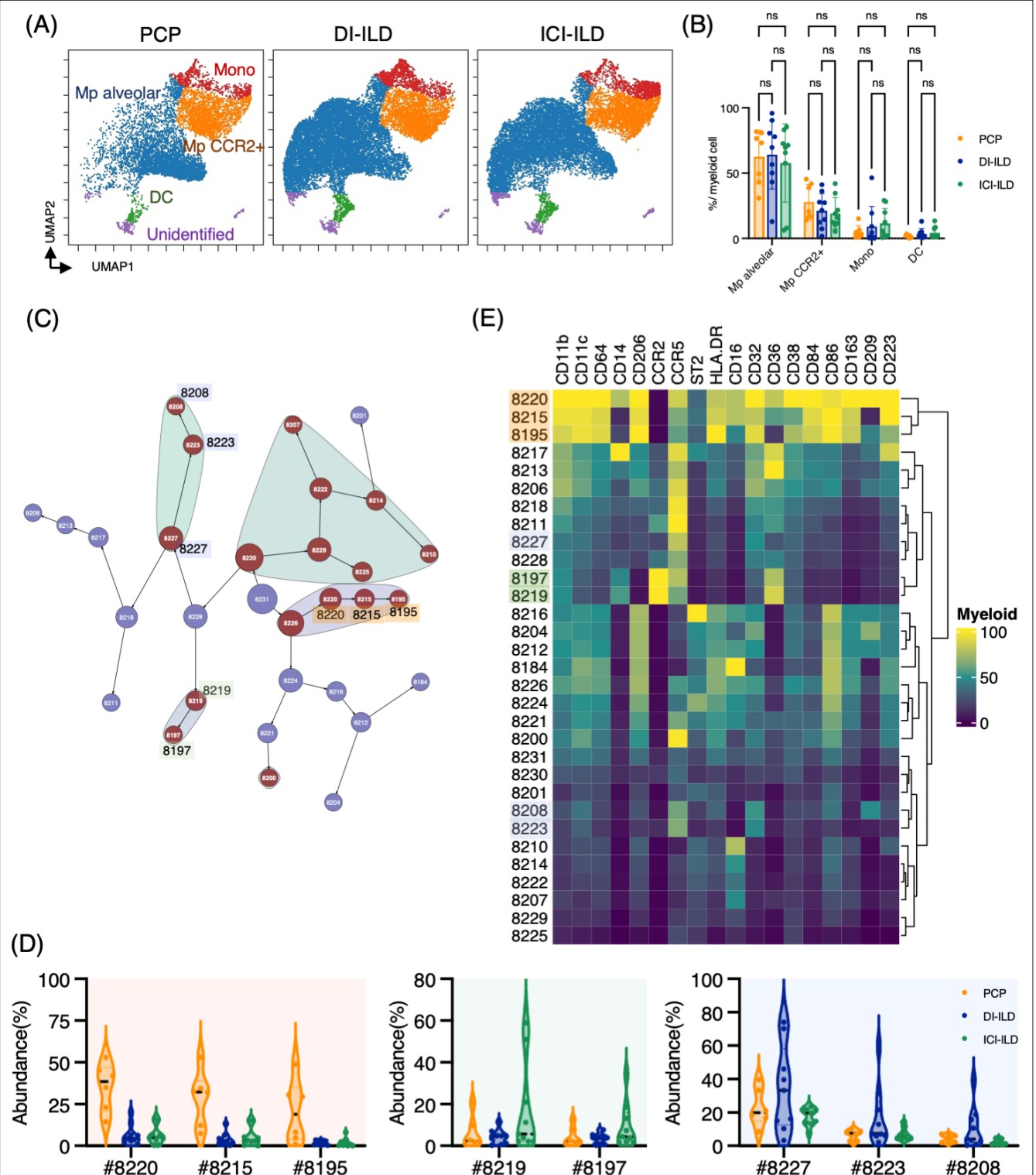

**Figure 3.** Characterization of myeloid cell subsets in bronchoalveolar lavage fluid (BALF) from patients with *Pneumocystis jirovecii* pneumonia (PCP), cytotoxic drug-related interstitial lung disease (DI-ILD), and immune-checkpoint inhibitor-related ILD (ICI-ILD). (**A**) Uniform Manifold Approximation and Projection (UMAP) of concatenated samples visualizing the distribution of myeloid cell subpopulations in CD3$^-$ CD11b$^+$ CD11c$^+$ gated myeloid cells in BALF from patients with PCP, DI-ILD, and ICI-ILD. Monocytes were defined by CD64$^+$ CD14$^+$, CCR2$^+$ macrophages (Mp) by CD64$^+$ CD14$^-$ CCR2$^+$, alveolar Mp by CD64$^+$ CD14$^-$CCR2$^-$ CD206$^+$, dendritic cells (DC) by CD64$^-$CD14$^-$CD206$^-$CD11c$^+$ HLA-DR$^+$, and unidentified subsets by CD64$^-$ CD11b$^{+/-}$ CD11c$^{+/-}$ CD14$^-$ CD206$^-$. (**B**) Percentage of myeloid cell subpopulations in PCP (n = 7), DI-ILD (n = 9), and ICI-ILD (n = 9). Dot plots represent individual samples.

*Figure 3 continued on next page*

*Figure 3 continued*

(**C**) Citrus network tree visualizing the hierarchical relationship of each marker between identified myeloid cell populations gated by CD45⁺ CD3⁻ CD11b⁺ CD11c⁺ from PCP (n = 6), DI-ILD (n = 9), and ICI-ILD (n = 9). Clusters with significant differences are represented in red, and those without significant differences in blue. Circle size reflects the number of cells within a given cluster. (**D**) Citrus-generated violin plots for six representative and differentially regulated populations. Each cluster number (#) corresponds to the number shown in panel (**C**). All differences in abundance were significant at a false discovery rate < 0.01. (**E**) Heatmap demonstrates the expression of various markers in different clusters of myeloid cells, as identified through the Citrus analysis.

The online version of this article includes the following figure supplement(s) for figure 3:

**Figure supplement 1.** The Citrus analysis of myeloid cell populations in bronchoalveolar lavage fluid (BALF) cells from *Pneumocystis jirovecii* pneumonia (PCP), cytotoxic drug-related interstitial lung disease (DI-ILD), and immune-checkpoint inhibitor-related ILD (ICI-ILD).

**Figure supplement 2.** Myeloid cell marker expression on the Uniform Manifold Approximation and Projection (UMAP) in *Figure 3A*.

## Marked CD16⁺ T cell expansion in BALF from a fatal case of PCP

One case of PCP that developed during the use of immunosuppressive drugs after living donor liver transplantation had a fatal course despite intensive treatment. Upon admission, chest CT images revealed the emergence of diffuse ground glass opacities and infiltration in the bilateral lungs (*Figure 5A*). The serum levels of β-D-glucan and KL-6 were highly elevated (1146 pg/mL and 1561 U/mL, respectively). The T cell percentage was 51.3%, and the CD4/CD8 ratio was 0.58, which was not particularly different from other PCP cases (*Figure 5B*). Strikingly, 97.5% of T cells expressed CD16 in the BALF from the fatal case (*Figure 5C and D*). Further, not only the proportion but the CD16 intensity was the highest in the fatal case among PCP cases (*Figure 5E*). On the contrary, there did not appear to be any significant differences in myeloid cell fractions, and CD16 expression was not particularly high in the fatal case (*Figure 5F*). The correlation matrix of clinical parameters and mass cytometry parameters revealed that CD16 intensity in T cells may be correlated with disease severity (*r* = 0.748, p=0.053) and β-D-glucan (*r* = 0.868, p=0.011) (*Figure 5G*). Therefore, CD16 expression in T cells is not only a characteristic of PCP but also a potential indicator of disease severity. For reference, we investigated CD16 expression in BALF T cells from healthy controls along with severe COVID-19 patients using the publicly available single-cell RNA-sequencing data (GSE145926). In the BALF of healthy controls, *FCGR3A* (CD16) expression was observed in 18.7% of CD4⁺ T cells and 7.22% of CD8⁺ T cells in healthy controls. In contrast, severe COVID-19 patients exhibited expression rates of 17.2% in CD4⁺ T cells and 22.1% in CD8⁺ T cells (*Figure 5H and I*). Notably, CD8⁺ T cells from COVID-19 patients showed significantly higher *FCGR3A* (CD16) expression compared to those from healthy controls (0.31 ± 0.62 vs 0.12 ± 0.43, p=2.47e-08), while no significant difference was observed in CD4⁺ T cells between the two groups (0.22 ± 0.53 vs 0.34 ± 0.73, p=0.407).

## Discussion

Here, we have demonstrated the characteristic immune cell subpopulations present in BALF from patients with PCP, DI-ILD, and ICI-ILD. Our analysis revealed an expansion of CD16⁺ T cells in patients with PCP, an increase in CD57⁺ CD8⁺ T cells expressing immune checkpoints, and FCRL5⁺ B cells in ICI-ILD.

CD16 is a low-affinity Fc receptor (FcγRIIIa) for IgG, commonly expressed on NK cells as well as on neutrophils and monocytes (*Ravetch and Bolland, 2001*), with only a small fraction of T cells expressing CD16 in healthy peripheral blood (*Sandor and Lynch, 1993*) and BALF (*Figure 5H and I*). On the contrary, CD16⁺ T cells are induced in specific conditions, such as chronic hepatitis C infection (*Björkström et al., 2008*) and COVID-19 (*Georg et al., 2022*). In chronic hepatitis C infection, CD16⁺ CD8 T cells displayed a late-stage effector phenotype with high levels of perforin with a restricted TCR profile (*Björkström et al., 2008*). Stimulation of CD16 on CD8 T cells evokes a vigorous response such as degranulation and cytokine production (IFN-γ and TNF-α) similar to that of CD16 stimulation in NK cells (*Björkström et al., 2008*). These CD16⁺ CD8⁺ T cells also express a senescent marker, CD57, which differs from our findings that expanding CD16⁺ CD8 T cells in PCP did not express CD57 (*Figure 2F*).

Georg et al. recently discovered that COVID-19 severity was associated with highly activated CD16⁺ T cells, which exhibit increased cytotoxic functions (*Georg et al., 2022*). Although this increase was observed in CD8⁺ T cells and not in CD4⁺ T cells, the augmented population of CD16⁺ T cells in

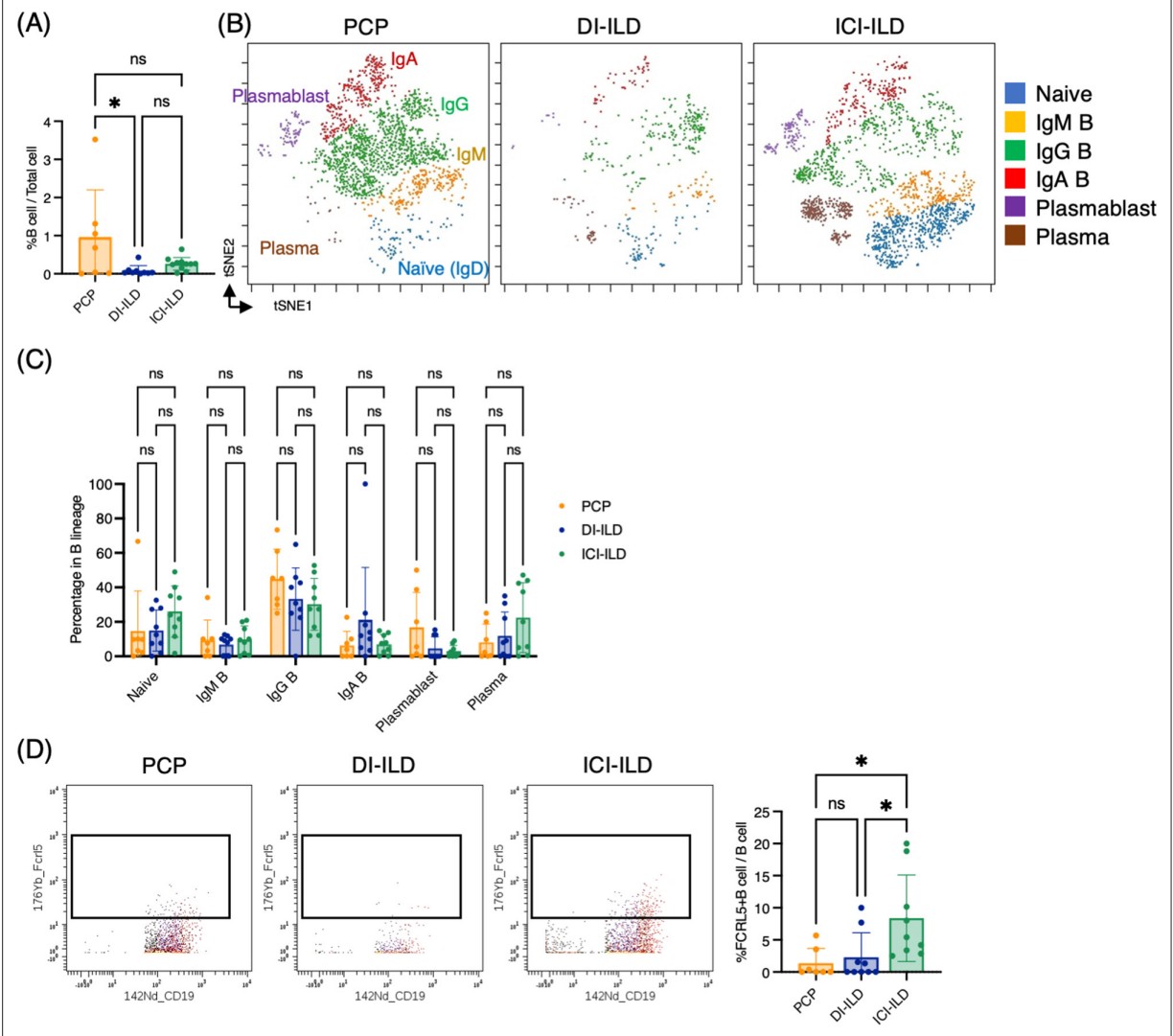

**Figure 4.** Characterization of B cell subsets in bronchoalveolar lavage fluid (BALF) from patients with *Pneumocystis jirovecii* pneumonia (PCP), cytotoxic drug-related interstitial lung disease (DI-ILD), and immune-checkpoint inhibitor-related ILD (ICI-ILD). (**A**) Percentages of B cells and plasma cells in CD45$^+$ BALF cells. (**B**) t-stochastic neighborhood embedding (t-SNE) plots of concatenated samples visualizing the distribution of B cell subpopulations in CD64$^-$CD3$^-$ and CD19$^+$ or CD138$^+$ gated B cells in BALF from patients with PCP, DI-ILD, and ICI-ILD. Naive B cells are defined by CD19$^+$IgD$^+$, IgM B cells: CD19$^+$ IgM$^+$, IgG B cells: CD19$^+$ IgG$^+$, IgA B cells: CD19$^+$ IgA$^+$, plasmablasts: CD19$^+$ CD27$^+$ CD38$^+$ CD138$^-$, plasma cells: CD19$^-$ CD138$^+$ and IgG$^+$ or IgA$^+$. (**C**) Percentages of B cell subpopulations in PCP (n = 7), DI-ILD (n = 9), and ICI-ILD (n = 9). Dot plots represent individual samples. (**D**) Two-dimensional dot plots depicting FCRL5-expressing B cells within a gated population of B cells defined as CD64$^-$CD3$^-$ and CD19$^+$ or CD138$^+$. Percentages of FCRL5-expressing B cells within the total B cell population are also shown.

The online version of this article includes the following figure supplement(s) for figure 4:

**Figure supplement 1.** B cell marker expression on the t-stochastic neighborhood embedding (t-SNE) projection in *Figure 4B*.

BALF from patients with severe COVID-19 was further substantiated by data mining of scRNA-seq (*Figure 5H and I*). CD16 expression allows T cells to degranulate and exert cytotoxicity in an immune-complex-mediated, TCR-independent manner. Moreover, CD16$^+$ T cells from COVID-19 patients caused injury to microvascular endothelial cells and induced the release of neutrophil and monocyte chemoattractants (*Georg et al., 2022*). Georg et al. also discovered that severe COVID-19 induced increased generation of C3a, which further activates CD16$^+$ cytotoxic T cells. The proportions of activated CD16$^+$ T cells and plasma levels of complement proteins upstream of C3a were found to be associated with fatal outcomes of COVID-19, proving support for the pathological role of exacerbated cytotoxicity and complement activation in COVID-19 (*Georg et al., 2022*). Given these findings and

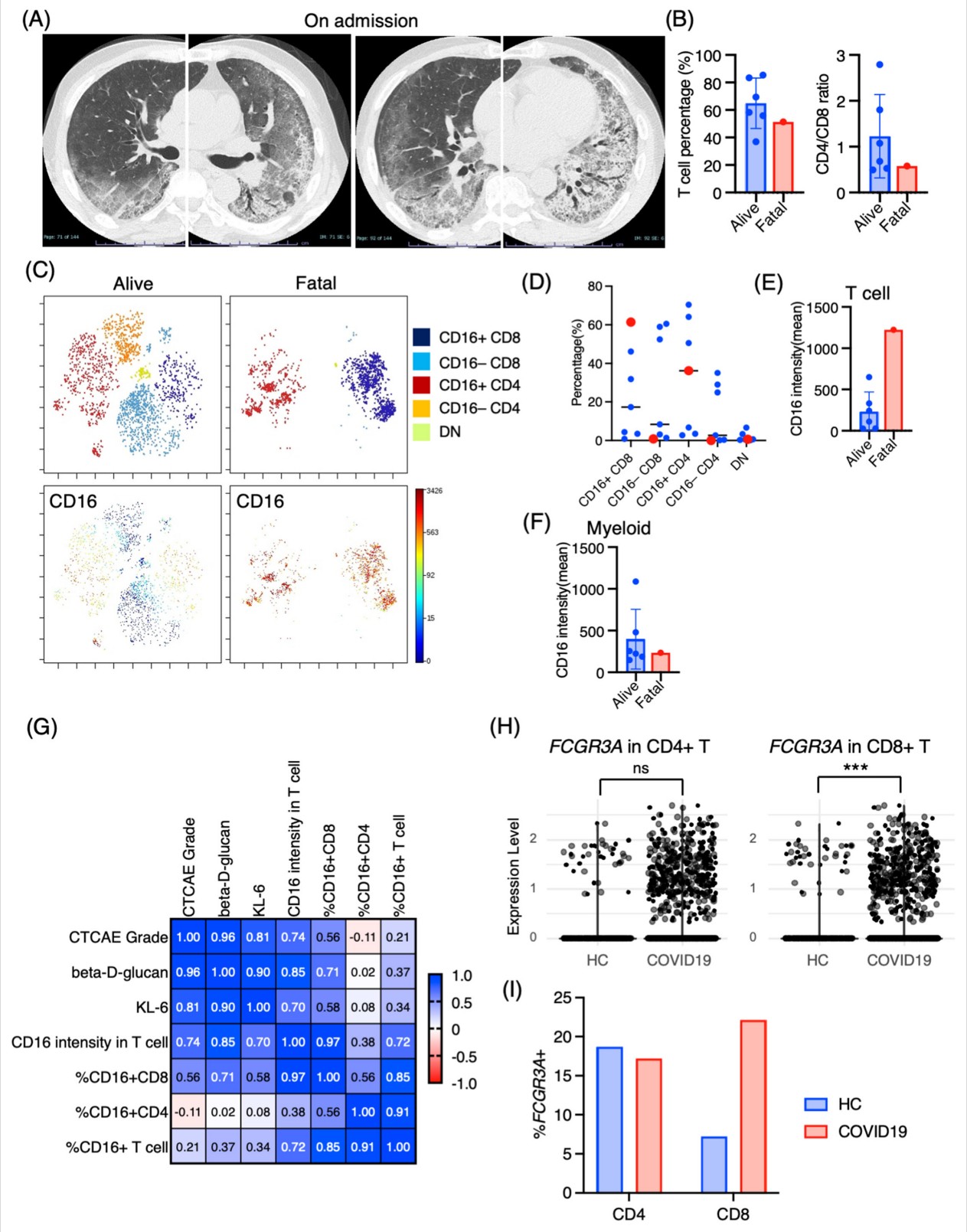

**Figure 5.** Immunological phenotypes in a fatal case of *Pneumocystis jirovecii* pneumonia (PCP). (**A**) Chest computed tomography images of the patient upon admission reveal the emergence of bilateral diffuse ground glass opacities and infiltration in both lungs. (**B**) Comparison of T cell percentage and CD4/CD8 ratio between a fatal case and surviving cases of PCP. (**C**) t-stochastic neighborhood embedding (t-SNE) plots illustrating the distribution of T cell subpopulations in bronchoalveolar lavage fluid (BALF) T cells (gated as CD45⁺CD2⁺CD3⁺) from a fatal case and surviving cases of PCP. Double-

*Figure 5 continued on next page*

*Figure 5 continued*

negative (DN) T cells were defined as CD4⁻CD8⁻ T cells. (**D**) Percentages of each T cell subpopulation. Red dots represent the value of the fatal case. (**E**) Mean CD16 intensity in the T cell population. (**F**) Mean CD16 intensity in the myeloid cell population. (**G**) The correlation matrix in PCP cases. Pearson *r* values are shown in each square. (**H**) Violin plots illustrate *FCGR3A* (CD16) expression intensity on CD4⁺ and CD8⁺ T cells, with each dot representing an individual cell in BALF from healthy controls (HCs) and COVID-19 patients, derived from single-cell RNA-seq dataset GSE145926. The significance of differences was assessed using the Wilcoxon test. ns: not significant, ***p<0.001. (**I**) The proportion of *FCGR3A* (CD16)-positive cells within the CD4⁺ and CD8⁺ T cell populations in BALF from HC and COVID-19 patients, with the *FCGR3A* expression threshold set at 0.5.

The online version of this article includes the following figure supplement(s) for figure 5:

**Figure supplement 1.** The correlation matrix with the CTCAE and intensity of each T cell markers in *Pneumocystis jirovecii* pneumonia (PCP) cases.

---

the observation of the highest proportion of CD16⁺ T cells in the fatal case of PCP, we hypothesize that these CD16⁺ T cells may possess excessive cytotoxicity toward pulmonary microvascular endothelial cells, contributing to lung injury in PCP, in addition to targeting *P. jirovecii*. Based on the therapeutic efficacy of an anti-complement therapy (anti-C5a antibody) for severe COVID-19 (*Campbell, 2020*; *Vlaar et al., 2020*), anti-complement treatments may have the potential to treat severe PCP. One of the other T cell markers correlated with the severity of PCP was CCR7, CD7, and CD57 (*Figure 2— figure supplement 2*), although the precise biological significance of the correlation remains to be elucidated.

We discovered an increase of CD57⁺ CD8⁺ T cells expressing immune checkpoints, PD-1⁺ TIGIT⁺ LAG3⁺ TIM-3⁺ in ICI-ILDs, which supports our previous findings of increased proportions of CD8⁺ T cells positive for both PD-1 and TIM-3 or TIGIT in ICI-ILD (*Suzuki et al., 2020*). In the 'Materials and methods' section, the low detection of PD-1 expression on T cells in patients treated with nivolumab was noted; this was due to the competitive nature of the PD-1 detection antibody EH12.2 with nivolumab (*Yanagihara et al., 2020*). The absence of a metal-conjugated PD-1 antibody with the MIH4 clone presented a limitation in our study. Ideally, we would have conjugated the MIH4 antibody with 155Gd for our analysis, which is a refinement we aim to incorporate in future research. A population of PD-1⁺ TIM-3⁺CD8⁺ cells was detected in the peritumoral pleural effusion and ILD lesions of a cancer patient undergoing ICI treatment (*Yanagihara et al., 2017*). Through the application of next-generation sequencing technology to the DNA encoding the complementarity-determining region of the T cell receptor (TCR), identical T cell clones were identified in both peritumoral pleural effusion and ILD lesions (*Tanaka et al., 2018*). Similar to ICI-ILD, a recent study identified T cells specific to α-myosin drive ICI-related myocarditis (*Axelrod et al., 2022*). These results indicate that ICI-activated CD8⁺ T cells, with a phenotype of PD-1⁺ TIGIT⁺ LAG3⁺ TIM-3⁺, could potentially trigger ILD by recognizing self-peptides or shared epitopes between tumors and lungs. CD16⁺ CD8⁺ T cells in PCP exhibit high CXCR3 and ST2 expression, and CD57⁺ CD8⁺ T cells expressing immune checkpoints in ICI-ILD express high CXCR3 (consistent with the previous study; *Kim et al., 2020*). Therefore, these CD8⁺ T cells could potentially migrate into the lungs via chemokines, such as CXCL4, CXCL9, CXCL10, and IL-33 (*Van Raemdonck et al., 2015*; *Griesenauer and Paczesny, 2017*).

Regarding CD57 expression in T cells, it is generally considered a senescent marker (*Fehlings et al., 2022*). However, CD57⁺ CD8⁺ T cells in the periphery show clonal expansion with the overlap of the TCR repertoire of tumor-infiltrating T cells with favorable response to ICI in cancer patients (*Fehlings et al., 2022*), indicating that these CD57⁺ CD8⁺ T cells may have a highly active phenotype in ICI-treated individuals.

FCRL5 is a B cell-restricted member of the Fc receptor-like family encoded by the *IRTA2* gene (*Rostamzadeh et al., 2018*). In both rheumatoid arthritis (*Owczarczyk et al., 2011*) and granulomatosis with polyangiitis and microscopic polyangiitis (*Owczarczyk et al., 2020*), increased expression of FCRL5 was indicative of a positive response to rituximab. Additionally, our recent research revealed an increase in FCLR5⁺ B cells in connective-tissue disease-related ILD (*Hata et al., 2023*). These findings suggest a potential pathological role of FCRL5 in autoimmunity. Given that ICI-ILD is considered a type of irAE, it is reasonable to hypothesize that FCRL5⁺ B cells may be one of the immune cells associated with autoimmunity.

Regarding myeloid cells, we found that CD14⁺ CCR2⁺ CCR5⁺ monocyte subpopulations were prevalent in ICI-ILD. *Franken et al., 2022* revealed a decrease in anti-inflammatory resident alveolar macrophages and an increase in pro-inflammatory 'M1-like' monocytes (expressing TNF, IL-1B, IL-6,

IL-23A, and GM-CSF receptor CSF2RA, CSF2RB) in BALF from ICI-ILD compared with controls. Taken together with this report, the CD14$^+$ CCR2$^+$ CCR5$^+$ monocytes we found may be an M1-like monocyte.

Our study has several limitations, including the absence of data from healthy individuals, a relatively small sample size, and a retrospective design that resulted in missing clinical data for certain cases. Selection bias may also be a factor as only patients who underwent bronchoalveolar lavage were eligible for enrollment in this study. It is important to note that these findings demonstrate a correlation rather than proof of causation.

In summary, our study has shown distinct immune cell phenotypes in PCP, DI-ILD, and ICI-ILD. Specifically, we have identified an expansion of CD16$^+$ T cells in PCP, as well as an increase in CD57$^+$ CD8$^+$ T cells expressing immune checkpoints, FCRL5$^+$ B cells, and CCR2$^+$ CCR5$^+$ monocytes in ICI-ILD, which may play a pathogenic role. Based on these findings, further confirmatory research may lead to the development of diagnostic methods and novel strategies that target these specific cell populations to treat chemotherapy-induced pneumonitis.

## Materials and methods

### Patients

In this retrospective study, patients who were newly diagnosed with PCP, DI-ILD, and ICI-ILD and had undergone BALF collection at Kyushu University Hospital from January 2017 to April 2022 were included. The retrospective study was approved by the Ethics Committee of Kyushu University Hospital (reference number 22117-00). Patient consent was waived due to an opt-out method that was approved by the Ethics Committee of Kyushu University Hospital. Diagnostic criteria for PCP, DI-ILD, and ICI-ILD were in accordance with those previously described (*Skeoch et al., 2018*; *Asai et al., 2022*; *Apostolopoulou and Fishman, 2022*; *Delaunay et al., 2019*). A visual representation of the experimental and analytical workflow can be found in *Figure 6*.

### Mass cytometry

Metal-tagged antibodies were obtained from Standard Biotools or purchased in a purified form (*Supplementary file 1*) and then labeled with metals using the Maxpar Antibody Labeling Kit (Standard Biotools) as instructed by the manufacturer. The cell labeling was performed as previously described (*Hata et al., 2023*). Upon collection, BALF samples were immediately centrifuged at 300 × *g* for 5 min to pellet the cells. The resultant cell pellets were then resuspended in Cellbanker 1 cryopreservation solution (Takara, Cat# 210409). This suspension was aliquoted into cryovials and gradually frozen to –80°C and stored at –80°C until required for experimental analysis. Cryopreserved BALF cells were thawed in PBS and then stained with Cell-ID Cisplatin-198Pt (Standard Biotools #201198, 1:2000 dilution) in PBS before being incubated with FcR blocking reagent (Myltenyi, #130-059-901) and barcoded with each metal-labeled CD45 antibody (*Supplementary file 1*). After washing, CD45-labeled cells were mixed and stained with APC-conjugated FCRL5 antibodies (for panel #2), followed by staining with the antibody cocktail (panels #1 and #2, see *Supplementary file 1*). The antibody amount was determined through preliminary experiments with metal minus one. Cells were then washed, fixed with 1.6% formaldehyde, and resuspended in Cell-ID Intercalator 103Rh (Standard Biotools #201103A) in Fix and Perm buffer (Standard Biotools) at 4°C overnight. For acquisition, cells were resuspended in MaxPar Cell Acquisition Solution (Standard Biotools #201240) containing one-fifth EQ Four Element Calibration Beads (Standard Biotools #201078) and acquired at a rate of 200–300 events/s on a Helios mass cytometer (Standard Biotools). Files were converted to FCS, randomized, and normalized for EQ bead intensity using the Helios software. Concatenating fcs files in the same group into one file was conducted by FlowJo v10.8 (BD Biosciences). Cytobank Premium (Cytobank Inc) was used to perform manual gating, visualization of t-distributed stochastic neighbor embedding (viSNE), UMAP analysis, and Citrus analysis (*Bruggner et al., 2014*).

### Data analysis

Cisplatin-positive cells and doublets were excluded to select live cells, and CD45$^+$ cells were further analyzed. For T cells, CD2$^+$ CD3$^+$ cells were gated and subjected to UMAP and Citrus algorithms. The UMAP analysis included clustering channels for CD4, CD8a, CD27, CD28, CD45RA, CD45RO, Fas, and used the parameters: numbers of neighbors = 15, minimum distance = 0.01. The Citrus

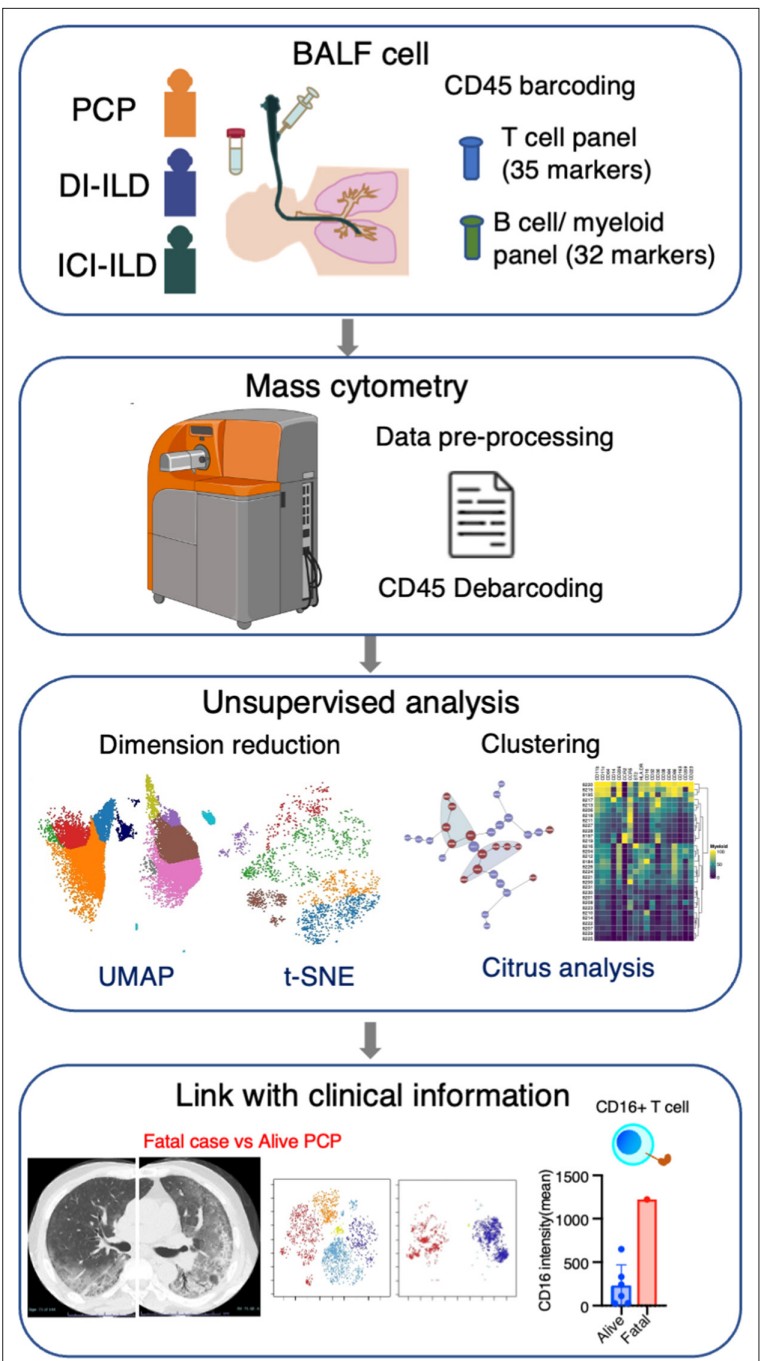

**Figure 6.** Graphical abstract of the study. Bronchoalveolar lavage fluid (BALF) samples were collected from patients with *Pneumocystis jirovecii* pneumonia (PCP), cytotoxic drug-induced interstitial lung disease (DI-ILD), and immune-checkpoint inhibitor-related ILD (ICI-ILD). Subsequently, BALF cells were analyzed using mass cytometry with a T cell panel (35 markers) and B cell/myeloid cell panel (32 markers) following CD45 barcoding for sample identification. The study found that there was a significant increase in the expansion of CD16+ T cells in patients with PCP, with the highest CD16 expression observed in a fatal case of PCP.

algorithm for T cells included clustering channels for CD4, CD5, CD7, CD8a, CD11a, CD16, CD27, CD28, CD44, CD45RA, CD45RO, CD49d, CD57, CD69, CD226, Fas, IL-2R, PD-L1, PD-L2, PD-1, OX40, TIGIT, TIM3, CTLA-4, CD223 (LAG-3), BTLA, ICOS, ST2, CCR7, CXCR3, HLA-DR, and used the parameters: association models = nearest shrunken centroid (PAMR), cluster characterization = abundance, minimum cluster size = 5%, cross-validation folds = 5, false discovery rate (FDR) = 1%. The Citrus

analysis excluded nivolumab-induced ILD cases due to the apparent loss of PD-1 detection caused by competitive inhibition by nivolumab (*Figure 2—figure supplement 1B*; *Yanagihara et al., 2020*). The viSNE analysis for T cells included clustering channels for CD4, CD5, CD7, CD8a, CD11a, CD16, CD27, CD28, CD44, CD45RA, CD45RO, CD49d, CD57, CD69, CD226, Fas, IL-2R, PD-L1, PD-L2, PD-1, OX40, TIGIT, TIM3, CTLA-4, CD223 (LAG-3), BTLA, ICOS, ST2, CCR7, CXCR3, and HLA-DR, and used the parameters: iterations = 1000,, perplexity = 30, theta = 0.5.

For myeloid cells, CD3⁻ CD11b⁺ CD11c⁺ cells were gated, and UMAP and Citrus algorithms were used. The UMAP analysis included clustering channels for CD11b, CD11c, CD64, CD14, CD16, CD206, HLA-DR, and CCR2, and used the parameters: numbers of neighbors = 10, minimum distance = 0.01. The Citrus algorithm included clustering channels for CD11b, CD11c, CD64, CD14, CD16, CD32, CD36, CD38, CD84, CD86, CD163, CD206, CD209, CD223, HLA-DR, CCR2, CCR5, and ST2, and used the parameters: association models = nearest shrunken centroid (PAMR), cluster characterization = abundance, minimum cluster size = 5%, cross validation folds = 5, FDR = 1%. One case of PCP was excluded from the Citrus analysis due to low cell numbers.

B cells and plasma cells were identified using gating of CD3⁻CD64⁻ and CD19⁺ or CD138⁺ cells. viSNE analysis was performed to cluster B cells using the following markers: CD19, CD38, CD11c, IgA, IgG, CD138, CD21, ST2, CXCR5, CD24, CD27, TIM-1, IgM, HLA-DR, IgD, and FCRL5. The analysis was performed on both individual and concatenated files using the parameters of 1000 iterations, perplexity of 30, and theta of 0.5.

The selection of the dimensionality reduction technique, UMAP or viSNE, was made based on their ability to retain the relationships between global structures and the distances between cell clusters (where UMAP outperformed viSNE) and their ability to present a distinct and non-overlapping portrayal of cell subpopulations, facilitating the identification of inter-group variations (where viSNE performed better than UMAP).

## Statistical analysis

We utilized a PAMR association model with a stringent threshold of 1% FDR, as described in the 'Data analysis' section, for the Citrus algorithm experiment. Student's two-tailed unpaired *t*-test was employed to conduct a comparative analysis between the two groups. To determine the significance of serum β-D-glucan and KL-6 among three groups, a statistical analysis was performed using Kruskal–Wallis tests. We evaluated the manually gated cell proportions by performing a two-way ANOVA, along with Tukey's multiple comparison tests. Data were analyzed using GraphPad Prism 9 software. Statistical significance was considered to be achieved when the p-value was <0.05.

## Analysis of single-cell RNA sequencing data of human BALF cells

For our investigation, we utilized the publicly available dataset GSE145926, which contains single-cell RNA-sequencing data from BALF cells of both healthy controls (n = 3) and patients with severe COVID-19 (n = 6) (*Liao et al., 2020*). We initiated our analysis by normalizing the filtered gene-barcode matrix using the 'NormalizeData' function in Seurat version 4, applying the default settings. Subsequently, we identified the top 2000 variable genes utilizing the 'vst' method available in the 'FindVariableFeatures' function of Seurat. Principal component analysis and UMAP were then conducted to reduce dimensionality and visualize the data. To isolate T cells for further analysis, we defined them as cells expressing *CD2* and *CD3E* with expression values greater than 1. We specifically examined the expression of *FCGR3A* (CD16) by setting an expression threshold of 0.5 to identify relevant cell populations. The resulting expression patterns were illustrated using violin plots and bar plots, which were generated using the 'ggplot' function in R.

## Acknowledgements

We thank Ms. Sanae Sekihara, PhD, from Standard Biotools, for her technical assistance. We also extend our appreciation to the Medical Research Center Initiative for High Depth Omics at Kyushu University. This research was supported by the Kakihara Foundation, Boehringer Ingelheim (TY), and the Japan Agency for Medical Research and Development (YF).

## Additional information

### Competing interests

Toyoshi Yanagihara: This research was supported by the Kakihara Foundation and Boehringer Ingelheim. The other authors declare that no competing interests exist.

### Funding

| Funder | Grant reference number | Author |
|---|---|---|
| The Kakihara Foundation | 4-9-10 | Toyoshi Yanagihara |
| Boehringer Ingelheim Japan | | Toyoshi Yanagihara |
| Japan Agency for Medical Research and Development | | Yoshinori Fukui |

The funders had no role in study design, data collection and interpretation, or the decision to submit the work for publication.

### Author contributions

Toyoshi Yanagihara, Conceptualization, Resources, Data curation, Formal analysis, Funding acquisition, Investigation, Visualization, Methodology, Writing – original draft, Project administration; Kentaro Hata, Formal analysis, Investigation, Visualization, Methodology, Writing - review and editing; Keisuke Matsubara, Data curation, Formal analysis, Investigation, Visualization, Methodology, Writing - review and editing; Kazufumi Kunimura, Formal analysis, Supervision, Investigation, Methodology, Writing - review and editing; Kunihiro Suzuki, Resources, Data curation, Methodology, Writing - review and editing; Kazuya Tsubouchi, Investigation, Methodology, Writing - review and editing; Satoshi Ikegame, Supervision, Investigation, Methodology, Writing - review and editing; Yoshihiro Baba, Supervision, Investigation, Visualization, Methodology, Writing - review and editing; Yoshinori Fukui, Supervision, Visualization, Methodology, Writing - review and editing; Isamu Okamoto, Supervision, Methodology, Writing - review and editing

### Author ORCIDs

Toyoshi Yanagihara ⓘ https://orcid.org/0000-0002-2212-0631
Kentaro Hata ⓘ http://orcid.org/0009-0001-9353-8381
Kazufumi Kunimura ⓘ http://orcid.org/0000-0002-3445-804X

### Ethics

The retrospective study was approved by the Ethics Committee of Kyushu University Hospital (reference number 22117-00). Patient consent was waived due to an opt-out method that was approved by the Ethics Committee of Kyushu University Hospital. Diagnostic criteria for PCP, DI-ILD, and ICI-ILD were in accordance with those previously described (Skeoch et al., 2018), (Asai et al., 2022), (Apostolopoulou and Fishman, 2022), (Delaunay et al., 2019).

Reviewer #2 (Public Review): https://doi.org/10.7554/eLife.87288.4.sa1
Reviewer #3 (Public Review): https://doi.org/10.7554/eLife.87288.4.sa2
Author response https://doi.org/10.7554/eLife.87288.4.sa3

## Additional files

### Supplementary files
• MDAR checklist
• Supplementary file 1. Mass cytometry antibody panels.

## Data availability

Raw fcs files, the case list and the code for scRNA-seq are available at https://doi.org/10.5061/dryad.2ngf1vhxd.

The following dataset was generated:

| Author(s) | Year | Dataset title | Dataset URL | Database and Identifier |
|---|---|---|---|---|
| Yanagihara T, Hata K, Matsubara K, Kunimura K, Suzuki K, Tsubouchi K, Ikegame S, Baba Y, Fukui Y, Okamoto I | 2024 | Exploratory mass cytometry analysis reveals immunophenotypes of cancer treatment-related pneumonitis | https://doi.org/10.5061/dryad.2ngf1vhxd | Dryad Digital Repository, 10.5061/dryad.2ngf1vhxd |

The following previously published dataset was used:

| Author(s) | Year | Dataset title | Dataset URL | Database and Identifier |
|---|---|---|---|---|
| Liao M, Liu Y, Yuan J, Wen Y, Xu G, Zhao J, Cheng L, Li J, Wang X, Wang F, Liu L, Amit I, Zhang S, Zhang Z | 2020 | Single-cell landscape of bronchoalveolar immune cells in COVID-19 patients | https://www.ncbi.nlm.nih.gov/geo/query/acc.cgi?acc=GSE145926 | NCBI Gene Expression Omnibus, GSE145926 |

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
