## [Editor Report · eLife assessment]

This study presents a **valuable** inventory of immune signatures that are correlated with cancer treatment-related pneumonitis. The data were collected and analyzed using validated methodology and can be used as a starting point for further prospective studies. The authors have provided an scRNA-seq analysis with an HD baseline using publicly available dataset and the evidence for their claims is **convincing**.

---

## [Referee Report · Reviewer #2 (Public Review)]

Yanagihara and colleagues investigated the immune cell composition of bronchoalveolar lavage fluid (BALF) samples in a cohort of patients with malignancy undergoing chemotherapy and with lung adverse reactions including Pneumocystis jirovecii pneumonia (PCP) and immune-checkpoint inhibitors (ICIs) or cytotoxic drug induced interstitial lung diseases (ILDs). Using mass cytometry, their aim was to characterize the cellular and molecular changes in BAL to improve our understanding of their pathogenesis and identify potential biomarkers and therapeutic targets. In this regard, the authors identify a correlation between CD16 expression in T cells and the severity of PCP and an increased infiltration of CD57+ CD8+ T cells expressing immune checkpoints and FCLR5+ B cells in ICI-ILD patients.

The conclusions of this paper are mostly well supported by data, but some aspects of the data analysis need to be clarified and extended.

The authors should elaborate on why different sets of markers were selected for each analysis step. E.g., Different sets of markers were used for UMAP, CITRUS and viSNE in the T cell and myeloid analysis.

---

## [Referee Report · Reviewer #3 (Public Review)]

The authors collected BALF samples from lung cancer patients newly diagnosed with PCP, DI-ILD or ICI-ILD. CyTOF was performed on these samples, using two different panels (T-cell and B-cell/myeloid cell panels). Results were collected, cleaned-up, manually gated and pre-processed prior to visualisation with manifold learning approaches t-SNE (in the form of viSNE) or UMAP, and analysed by CITRUS (hierarchical clustering followed by feature selection and regression) for population identification - all using Cytobank implementation - in an attempt to identify possible biomarkers for these disease states. By comparing cell abundances from CITRUS results and qualitative inspection of a small number of marker expressions, the authors claimed to have identified an expansion of CD16+ T-cell population in PCP cases and an increase in CD57+ CD8+ T-cells, FCRL5+ B-cells and CCR2+ CCR5+ CD14+ monocytes in ICI-ILD cases.

By the authors' own admission, there is an absence of healthy donor samples and, perhaps as a result of retrospective experimental design and practical clinical reasons, also an absence of pre-treatment samples. The entire analysis effectively compares three yet-established disease states with no common baseline - what really constitutes a "biomarker" in such cases? These are very limited comparisons among three, and only these three, states.

By including a new scRNA-Seq analysis using a publicly available dataset, the authors addressed this fundamental problem. Though a more thorough and numerical analysis would be appreciated for a deeper and more impactful analysis, this is adequate for the intended objectives of the study.

---

## [Author Response]

The following is the authors’ response to the previous reviews.

**eLife assessment**
This study presents a useful inventory of immune signatures that are correlated with cancer treatment-related pneumonitis. The data were collected and analysed using solid and validated methodology and can be used as a starting point for further functional studies.

We sincerely thank the editor for their encouraging comments regarding our study. As rightly pointed out, this study indeed serves as a pivotal starting point for subsequent functional studies.

**Reviewer #2 (Recommendations For The Authors):**
I greatly appreciate the authors diligence in addressing all the suggested points. The paper now presents significantly stronger evidence to support the findings.I do have one final question: Could you clarify how the correlation presented in Supplementary Figure 3 was calculated? Is it a Pearson correlation of CTCAE grade directly to marker expression? Additionally, could you explain how the significance was determined? The authors mention a significant correlation for CCR7, but the heatmap displays similarly high values for CD7 and CD57. Finally, I'm curious about the absence of CD16 in the heatmap.

Thank you for your insightful query. To clarify, the correlation shown in Supplementary Figure 3 was indeed calculated using the Pearson correlation coefficient. This involved correlating the CTCAE grade directly with the mean expression levels of each marker. The computations were conducted using GraphPad Prism version 9. Regarding the statistical significance, we defined a threshold of P < 0.05 as significant. Specifically, the P-values for CCR7, CD7, and CD57 were found to be 0.009, 0.035, and 0.039, respectively. Hence, while CCR7 showed a significant correlation, CD7 and CD57 also exhibited relatively high values, as correctly observed. We have added CD7 and CD57 along with CCR7 in the discussion section, though not to mention much for better focusing on CD16.

CD16 was initially omitted from Supplementary Figure 3 to prevent redundancy and preserve data clarity. Nonetheless, in light of your query, we have included CD16 in the correlation matrix to provide a comprehensive view of its association with other markers.

We hope this adequately addresses your question and further clarifies our findings.

**Reviewer #3 (Recommendations For The Authors):**
General suggestions for presentation in the future:It is essential to concretely define the numbers presented in all figures and plots. For example, in Figure 6 (I), what does it mean by "percentage representation of FCGR3A (CD16)"? Percentage of what? How did you calculate that? It is also important to show more statistics in general, for example, in dot plots like Figure 6 (H), where are the means and p-values? Little things like that completely change the impact of the figures. For the narrative of this paper, it is OK, but in the future, fine-tuning the presentation would massively improve the impact of the work which the contents deserve.

Thank you for your insightful feedback. Addressing your concerns, I have revised Figure 6H and Figure 6I to provide a more precise and informative presentation of our data. In Figure 6H, the violin plots illustrate the expression intensity of FCGR3A (CD16) on CD4+ and CD8+ T cells. Each dot represents an individual cell within the BALF from both healthy controls (HC) and COVID-19 patients. This data was derived from the single-cell RNA-seq dataset GSE145926. To enhance clarity and statistical robustness, I have now included p-values directly in Figure 6H. Additionally, for a more comprehensive understanding, the means ± standard deviation (SD) have been incorporated into the main text of the manuscript.

Regarding Figure 6I, it depicts the proportion of FCGR3A (CD16)-positive cells within the CD4+ and CD8+ T cell populations in BALF from HC and COVID-19 patients. The threshold for FCGR3A expression was set at 0.5. Upon further review and in response to your feedback, I realized an error in the calculation of the proportion of FCGR3A-positive cells among CD4+ and CD8+ T cells. Initially, the proportion of FCGR3A-positive CD4+ T cells was calculated in relation to the entire CD4+ T cell population, without differentiation between the groups. This has now been corrected, and the adjusted figures are presented in Figure 6I.

I am grateful for the opportunity to refine these figures, as your suggestions have not only helped to correct the error but have also significantly enhanced the impact and clarity of our work. Your guidance has been instrumental in improving the overall quality and presentation of our research, ensuring that the findings are communicated effectively and accurately.